# Clinical Update of Severe Fever with Thrombocytopenia Syndrome

**DOI:** 10.3390/v13071213

**Published:** 2021-06-23

**Authors:** Jun-Won Seo, Dayoung Kim, Nara Yun, Dong-Min Kim

**Affiliations:** Department of Internal Medicine, Division of Infectious Disease, College of Medicine, Chosun University, 588 Seosuk-dong, Dong-gu, Gwangju 501-717, Korea; jwseo83@chosun.ac.kr (J.-W.S.); dayz02@hanmail.net (D.K.); shine-0222@hanmail.net (N.Y.)

**Keywords:** severe fever with thrombocytopenia, SFTS virus, treatment

## Abstract

Severe fever with thrombocytopenia syndrome (SFTS) is an acute febrile illness characterized by fever, leukopenia, thrombocytopenia, and gastrointestinal symptoms such as diarrhea, nausea, and vomiting resulting from infection with the SFTS virus (SFTSV). The SFTSV is transmitted to humans by tick bites, primarily from *Haemaphysalis longicornis*, *Amblyomma testudinarium**, Ixodes nipponensis*, and *Rhipicephalus microplus*. Human-to-human transmission has also been reported. Since the first report of an SFTS patient in China, the number of patients has also been increasing. The mortality rate of patients with SFTS remains high because the disease can quickly lead to death through multiple organ failure. In particular, an average fatality rate of approximately 20% has been reported for SFTS patients, and no treatment strategy has been established. Therefore, effective antiviral agents and vaccines are required. Here, we aim to review the epidemiology, clinical manifestations, laboratory diagnosis, and various specific treatments (i.e., antiviral agents, steroids, intravenous immunoglobulin, and plasma exchange) that have been tested to help to cope with the disease.

## 1. Introduction

Severe fever with thrombocytopenia syndrome (SFTS) is an infectious zoonosis caused by the SFTS virus (SFTSV), a newly identified member of the genus Phlebovirus in the family Bunyaviridae carried by ticks (*Haemophysalis longicornis, Amblyomma testudinarium**, Ixodes nipponensis*, and *Rhipicephalus microplus*) (Figure 1). SFTS is characterized by fever, thrombocytopenia, and leukopenia [1,2,3].

SFTSV particles have a diameter of 80–120 nm and large (L), medium (M), and small (S) negative-sense RNA genomic segments [2]. The L segment encodes the 2084 amino acids of the viral RNA-dependent RNA polymerase (RdRp), which triggers replication and transcription of viral RNA. The M segment encodes the 1073 amino acids of the precursor of the glycoproteins Gn and Gc, which are responsible for the formation of viral particles and their attachment to target cells. Finally, the S segment encodes the nucleocapsid protein (N) and a nonstructural protein (NSs) via ambisense transcription [4]. According to phylogenetic analysis, SFTSVs can be classified into six genotypes (A–F) [5].

SFTSV released into the human body by SFTSV-infected ticks that attach to human skin and suck blood result in SFTS [2,6]. Therefore, SFTS generally occurs in individuals with plentiful contact opportunities with ticks. SFTS is also infectious to hospital workers who have been in contact with patients with severe SFTS at medical institutions and to family members living with SFTS patients [7,8,9].

Since the first report of SFTS in China in 2011, the disease has been recorded in South Korea [10] and Japan [11]. The number of patients with SFTS in East Asian countries has increased each year; however, patients are being treated using conservative treatment strategies due to the lack of effective treatment options. Thus, mortality rates remain high [12,13]. In this review, we describe the characteristics of SFTS and discuss the latest treatment strategies being tested to help to cope with the disease.

## 2. SFTSV Epidemiology and Transmission

SFTS has been reported in East Asia (central and eastern China, rural areas of South Korea, and western Japan). After the appearance of patients presenting with high fever, gastrointestinal symptoms, thrombocytopenia, leukopenia, and multiple organ failure (MOF) in March–July 2009 and after 2 years of epidemiological investigation in central and northeastern China (Jiangsu, Anhui, Hubei, Henan, Shandong, and Liaoning), SFTSV was first identified in 2011 [2]. The first confirmed case of SFTS was discovered in Japan at the end of 2012, and epidemiological surveys from 2013 to 2017 showed that SFTS patients had an average age of 74 years and lived primarily in eastern and southern Japan [14]. In South Korea, 1203 patients (including 231 fatal cases) were reported between 2013 (when the first case of SFTS was reported) and August 2020 [15]. Most patients fell ill between April and November. Recently, patients with SFTS have also been identified in Southeast Asia (e.g., Vietnam, Pakistan, and Taiwan) in addition to three representative East Asian countries (China, Korea, and Japan) [16,17,18]. The study conducted by Tran et al. in Vietnam was a retrospective analysis of serum samples from 80 patients with acute febrile illness; the previous presence of SFTSV in Vietnam cannot be ruled out. In contrast, SFTSV was confirmed in animals in Taiwan after the aforementioned cases in humans, providing a stronger basis for the occurrence of SFTS [19]. In northwestern Missouri, United States, two patients with an SFTS-like illness were reported and underwent a clinical progress similar to that of SFTS. The newly discovered causative pathogen called the Heartland virus was most closely related to SFTSV genetically; nucleoprotein, nonstructural protein, glycoprotein, and polymerase sequencing indicated *Amblyomma*
*americanum*, which accounts for 99.9% of ticks in the region, as a potential vector [20,21].

Three out of six genotypes (F, A, and D) were dominant in China, and genotype B was dominant in South Korea and Japan [5,22]. Yun et al. performed genetic and phylogenetic analyses of serum or cerebrospinal fluid (CSF) samples obtained from patients with suspected SFTS in South Korea; the genotype B-2 strains were the most prevalent and associated with the highest mortality (43.8%), followed by strains of genotypes A and F [23]. The authors showed that different genotypes of SFTSV triggered different clinical manifestations in a ferret model. In particular, young adult ferrets infected with any of the genotypes did not experience weight loss and showed mildly elevated body temperature and short periods of viremia compared with aged ferrets. Aged ferrets infected with genotype B and D strains showed high fever, significant body weight loss, and a high viral load in the blood.

The seroprevalence of specific antibodies against SFTSV in the general population has been investigated in various epidemiological studies. A meta-analysis of 21 studies conducted in 7 provinces in central and eastern China showed that the seroprevalence of total antibodies (immunoglobulin G (IgG) and immunoglobulin M (IgM)) against SFTSV was 4.3%, with no differences according to age or sex [24]. However, two studies published in Japan showed that the seroprevalence of SFTSV antibodies in healthy Japanese living in SFTS-endemic regions of Japan was only 0.2–0.3% — a much lower seroprevalence than that detected in China [25,26]. In three rural areas of Korea, the seroprevalence of IgG antibodies against SFTSV was reported as 4.1%. A history of SFTS-related fever and gastrointestinal symptoms during the previous 3 years and a long career duration (≥41 years) were related to SFTSV seropositivity; however, age, number of family members, and number of tick bites during the previous year were not [27]. Among these three major East Asian countries where SFTS occurs, the seroprevalence of antibodies against SFTSV varies from country to country, showing that China and Korea have a high seroprevalence of antibodies against SFTSV, but Japan has a relatively low one. This may be a result of regional characteristics or may be due to differences in antibody detection methods.

SFTSV can be transmitted by various tick species, including *H. longicornis, A. testudinarium**, I. nipponensis*, and *R. microplus* [1,28,29]. The principal SFTS vector, *Haemaphysalis longicornis*, is primarily distributed in temperate regions and inhabits South Korea, Japan, China, Far East Russia, Australia, New Zealand, Fiji, New Caledonia, and Hawaii [30]. Kim et al. reported the detection of an identical SFTSV in a patient with SFTS encephalopathy and in the *H. longicornis* tick that bit her [31]. The case report suggests that SFTSV can be transmitted from *H. longicornis* to humans. The higher prevalence of *H. longicornis* in endemic regions than in non-endemic regions also suggests that it is the dominant SFTSV vector [1]. In addition, Zhuang et al. demonstrated the transovarial and transstadial transmission of SFTSV by detecting viral RNA in the ovary, salivary glands, and eggs of infected *H. longicornis* after microinjection of SFTSV into *H. longicornis* [32]. SFTSV is thought to circulate in a zoonotic cycle between ticks and vertebrates. Several animal serosurveys have identified SFTSV-specific antibodies in cattle, goats, sheep, dogs, pigs, and chickens [33,34,35,36,37]. Transmission from pets, such as cats or dogs, to humans has also been reported. For example, Kida et al. detected SFTSV in the serum of a veterinarian who had treated three sick cats, showing evidence that SFTSV was transmitted to the patient directly by a cat rather than a tick bite [38]. Moreover, Chung et al. reported possible SFTS transmission in dogs [39]. The patient diagnosed with SFTS stated that he had removed and crushed ticks from his dogs with his bare hands. One of the dogs had an elevated SFTS IgG titer according to immunofluorescence assay (IFA), amounting to 1:1024; however, SFTSV was not detected by PCR. The epidemiological findings of this case suggest a possible relationship between tick infestation in domestic dogs and SFTSV transmission to humans. In Japan, an epidemiological study on SFTS was conducted between 2013 and 2017, and 48% of the total patients confirmed close contact with companion animals within 2 weeks of the onset of the disease [40]. These findings show that contact precautions should be strictly followed by infected patients, even toward pets. Yu et al. reported that infectious SFTSVs could be shed through nasal discharge, saliva, and urine from SFTSV-infected ferrets. Furthermore, these authors suggested that urine specimens could be a source of SFTSV infection in humans by showing that ferrets in contact with SFTSV-positive urine specimens were infected with SFTSV [41]. Human-to-human transmission has been reported for patients in close contact with infected body fluids (e.g., upon needle-stick injury and during patient care) [7,8,9,42,43]. In 2014, four healthcare workers from a tertiary care hospital in Seoul, South Korea, who had contact with SFTS patients who underwent cardiopulmonary resuscitation were diagnosed with SFTS [9]. Nosocomial SFTSV transmission is also possible. Therefore, maintaining contact precautions when treating patients with suspected SFTS is important. Additionally, contaminated areas should be immediately cleaned with soap and water if damaged skin or mucous membranes are exposed to SFTS through patients’ blood, body fluids, secretions, or feces. In addition, the presence of SFTSV RNA has been reported in semen, suggesting that SFTSV is capable of sexual transmission [44]. Moon et al. found that SFTSV was transmitted from person to person through aerosols during aerosol-generating procedures. These authors suggested that standard, contact, and airborne precautions may be needed during aerosol-generating procedures [45]. Nevertheless, there is no definitive evidence for airborne transmission of SFTSV; therefore, further studies on this aspect are required.

## 3. Pathogenesis

SFTS pathogenesis has not been completely elucidated but is associated with a high viral load and inflammatory responses triggered by several cytokines, such as interleukin 1 receptor antagonist (IL-1RA), IL-6, IL-10, the granulocyte colony stimulating factor (G-CSF), and monocyte chemoattractant protein-1 (MCP-1) [46,47]. In addition, the SFTSV glycoproteins mediate receptor binding and viral entry into macrophages and dendritic cells and are targets for neutralizing humoral immunity [48]. Natural killer cells exert immunoregulatory functions through cytokine production [49,50]. Sun et al. discovered that the proportion of natural killer cells increases in the acute phase and in patients with severe SFTS, whereas that of CD3 and CD4 T lymphocytes is lower [51]. The increased proportion of natural killer cells leads to upregulation of the abovementioned inflammatory cytokines and ultimately triggers a cytokine storm. The inflammatory cytokines play an important role in the pathogenesis of viral diseases [46,49,52]. Unbalanced cytokine profiles have three distinct patterns. First, IL-1RA, IL-6, IL-10, G-CSF, and MCP-1 are more abundant in patients with severe SFTS, and IL-10 levels increase sharply in acute periods in both fatal and nonfatal cases and are low in healthy people or nonfatal cases [46]. Second, the SFTSV nonstructural protein encoded by the S segment (SFTSV-NSs) acts as an interferon (IFN) and an NF-κB antagonist and induces the expression of IL-10 to facilitate viral pathogenesis; the corresponding gene is an SFTSV virulence factor [53,54]. In contrast, the levels of the platelet-derived growth factor are regulated upon activation, normally T-expressed, and presumably secreted (RANTES) in low quantities that return to normal ranges during convalescence [46]. Lastly, the levels of IL-1β, IL-8, and macrophage inflammatory proteins 1α and 1β increased in severe SFTS cases and in survivors during convalescence [46]. These changes in cytokine levels are associated with viral load in the serum and play an important role in SFTS pathogenesis. In addition, the cytokines are associated with the clinical features. For example, MCP-1 causes progressive renal failure, inflammation, and fibrosis of the liver [55,56]; IL-8 participates in progressive renal failure and increases vascular permeability [55,57]. Moreover, the hemorrhagic fever symptoms of SFTS are caused by TNF-α, which leads to vasodilation and increased endothelial permeability [58]. Thrombocytopenia is caused by phagocytosis of splenic macrophages that scavenge the platelets to which the SFTSV is attached; such splenic clearance of virus-bound platelets is important for the pathogenesis of thrombocytopenia in patients with SFTS [59]. SFTSV replicates in various cell types in the body. Among these, infected monocytes avoid apoptosis and remain almost intact; therefore, the SFTSV within them can spread into the circulation via lymphatic drainage, causing viremia [53,60]. SFTSV-infected cells in lymph nodes are macrophages and class-switched B cells with an immunophenotype similar to that of plasmablasts [61]. In a surviving patient, fluorodeoxyglucose 18 positron emission tomography (^18^F-FDG PET) imaging revealed hypermetabolism in regional lymph nodes and the spleen [62]. The characteristic histologic findings of the lymph nodes include extensive necrosis and histiocytic proliferation [63].

## 4. Clinical Manifestations

The SFTS incubation period lasts approximately 5 to 14 days and can be affected by viral loads or the portal of entry for infection [64]. The usual skin marks on the tick bite do not present eschar, which is typical of patients with scrub typhus [65]. Most patients have fever, gastrointestinal manifestations (e.g., nausea, vomiting, abdominal pain, and diarrhea), and neurological symptoms such as altered mental status [2,64]. Laboratory findings relative to most SFTS patients revealed thrombocytopenia (<100,000/mm^3^) and leukopenia (<4000/mm^3^), accompanied by elevated alanine aminotransferase (ALT), aspartate aminotransferase (AST), and alkaline phosphatase (ALP) levels and acute kidney injury. In addition, lactate dehydrogenase (LDH) and ferritin levels also increase, and prolonged activated partial thromboplastin time (aPTT) and proteinuria with or without hematuria can be observed [2,64,66]. Chest radiographs in patients with SFTS primarily show cardiomegaly with or without pericardial effusion and patchy consolidation with ground-glass opacity (GGO), which helps in the early differentiation of SFTS from scrub typhus — which is characterized by interstitial pneumonia on chest radiographs [67].

Most patients with severe SFTS die within 2 weeks because of MOF, including acute kidney injury, myocarditis, arrhythmia, and meningoencephalitis during the second week of illness [68,69]. The average period from illness onset to death is 9 days [70]. The fatality rate of SFTS ranges from 6% to 21% [12,13,71]. Poor prognostic factors include advanced age, altered mental status, elevated serum LDH and AST levels, prolonged aPTT, and high viral RNA loads in the serum [13,71,72,73,74]. In particular, viral RNA load has been reported to provide relevant information for treatment plans or the prognosis of patients with SFTS, similar to cytokine, LDH, AST, and blood urea nitrogen (BUN) levels [75]. Yoshikawa et al. and Yang et al. analyzed the relationship between the number of SFTSV RNA copies determined by real-time PCR and the mortality rate of SFTS patients and found that deceased SFTS patients exhibited lower threshold cycle (Ct) values and, thus, a higher number of viral RNA copies than SFTS survivors [76,77]. In addition, advanced age is an important risk factor for a poor prognosis. Park et al. showed the effect of advanced age on SFTS prognosis using ferrets injected with SFTSV. Young ferrets (<2 years of age) did not show SFTS-related clinical symptoms or death; however, older ferrets (>4 years of age) exhibited severe bicytopenia (thrombocytopenia and leukopenia), high fever, high viral loads, and a 93% mortality rate [78]. These findings are consistent with observations in humans. Indeed, a multivariate analysis by Jeong et al. revealed that advanced age is a significant risk factor associated with high 30-day mortality in SFTS patients (adjusted hazard ratio (aHR), 1.10; 95% confidence interval (CI), 1.04–1.17) [79]. The occurrence of secondary infections in SFTS patients has also been reported and can be triggered by the presence of bacteria or fungi. Leukocytosis is occasionally observed in patients with SFTS, although most cases show leukopenia. If leukocytosis is observed, the possibility of a secondary infection should be considered. For example, Lee et al. identified a patient with fever and unstable vital signs; laboratory tests showed leukocytosis (white blood cell count, 24.83 × 10^3^) with thrombocytopenia (platelet count, 100,000/µL), proteinuria, and conjunctival hemorrhage; the patient suffered from SFTS and concomitant *Escherichia coli* bacteremia [80]. Bae et al. reported that among the 36% (16/45) of patients with SFTS who were admitted to an intensive care unit (ICU), 56% developed invasive pulmonary aspergillosis (IPA) within a median of 8 days. Moreover, SFTS case mortality was higher among patients with IPA than without IPA [81].

## 5. SFTS Diagnosis

SFTS is a disease that is difficult for medical staff to diagnose if they do not suspect it. Patients present with fever, low platelet counts, and white blood cell counts. SFTS should be suspected if the patients have a history of tick bites in endemic areas, such as central and eastern China, rural areas of South Korea, and southern Japan. Early diagnosis of SFTSV infection is important for patient survival. Laboratory confirmation is essential, because the clinical manifestations of SFTS are nonspecific; additionally, other tickborne diseases such as scrub typhus and anaplasmosis trigger similar symptoms [65,68]. Reverse transcriptase (RT) real-time PCR for the detection of viral RNA in the serum during the first week of illness is a highly sensitive and specific diagnostic tool for laboratory diagnosis of SFTS [82]. Viral RNA can be detected in the serum in the acute phase and until 20 days after symptom onset; however, testing serum samples within 2 weeks from the onset of illness is appropriate [82]. Due to considerable genetic differences among SFTSV residents, RT-PCR techniques based on the nucleotide sequence of SFTSV strains identified in China may be less susceptible to diagnoses of the SFTS lineage identified in other countries. Yoshikawa et al. developed a sensitive and specific conventional one-step RT-PCR method and quantitative one-step RT-PCR, which can detect both strains to overcome the aforementioned problem [76]. In addition, a variety of PCR techniques are being developed to easily and quickly diagnose SFTSV. Huang et al. developed a reverse transcription-loop-mediated isothermal amplification (RT-LAMP) technique to quickly identify new bunyavirus with 99% sensitivity and 100% specificity [83]. Baek et al. also showed that fast diagnosis within 30–60 min and sensitivity 10 times higher than that of conventional RT-PCR were possible with RT-LAMP [84]. IFA or an enzyme-linked immunosorbent assay (ELISA) are effective diagnostic methods for detecting viral-specific IgM and IgG in the serum 7 days after the onset of the disease; SFTS is diagnosed upon detection of IgM antibodies, observation of IgG antibody seroconversion, or at least a four-fold increase in antibody titer [64]. However, the IFA sensitivities for IgM and IgG detection within 2 weeks after the onset of symptoms are 32–62% and 63–76%, respectively; the IFA sensitivities of ELISA are 53–62% and 58–86%, respectively [85]. Therefore, IFA or ELISA may be insufficient for early SFTS diagnosis. Differential diagnosis includes viral diseases with hemorrhagic fever, such as hemorrhagic fever with renal syndrome (HFRS), severe dengue fever, thrombocytopenic purpura (TTP), leptospirosis, human granulocytic anaplasmosis (HGA), and Lyme disease. Patients with these diseases present clinical manifestations and laboratory results similar to those of patients with SFTS. Therefore, differential diagnosis is very important in areas where the diseases coexist with SFTS (e.g., South Korea, China, and Japan). In particular, scrub typhus and SFTS trigger similar clinical manifestations and laboratory findings in endemic areas. To distinguish the diseases, Kim et al. proposed a scoring system that showed 100% sensitivity and 97% specificity when a score ≥ 2 was obtained after the evaluation of four variables (i.e., altered mental status, leukopenia, prolonged aPTT, and normal C-reactive protein levels), which all weighed one point [65]. To exactly identify the diseases more easily and quickly, Li et al. suggested a multiplex real-time RT-PCR assay to conduct effective screening for early SFTS diagnosis and to differentiate it from other diseases (such as those provoked by the Hantaan, Seoul, and dengue viruses) in the acute phase [86]. Conversely, virus isolation for laboratory diagnosis is currently difficult to apply in the clinic because it should be performed in a biosafety level 3 (BSL 3) laboratory and takes approximately 2–5 days.

## 6. SFTS Treatment

No prospective randomized studies on treatment strategies have proven to be effective against SFTS. Conservative treatment, including hydration, transfusion, and administration of antipyretics, inotropic agents, and G-CSF is provided to alleviate symptoms such as fever, diarrhea, dehydration, bleeding tendency, and shock. However, rapidly progressing cases of the disease are difficult to properly treat in the acute stage; many patients with severe SFTS are believed to experience sepsis or septic shock due to confirmed MOF before being diagnosed. Therefore, early SFTS recognition is important. Various strategies have been attempted due to the difficulty of treatment and the high mortality rate of SFTS. In the next paragraphs, we introduce proposed treatment options.

### 6.1. Antiviral Agents

#### 6.1.1. Ribavirin

Several reports have demonstrated the effect of ribavirin on patients with diseases caused by bunyaviruses, such as HFRS and Crimean–Congo hemorrhagic fever [87,88]; the use of ribavirin to treat SFTS patients has also been examined. In 2017, Lee et al. published a study reporting the effects of ribavirin on SFTSV; in the in vitro study, ribavirin reduced SFTSV replication and cytopathic effects at a 50% inhibitory concentration (IC_50_) ranging from 3.69 to 8.72 µg/mL [89]. Li et al. conducted a prospective observational study of 2096 patients diagnosed with SFTS from April 2011 to October 2017 and found that case fatality decreased from 6.25% to 1.16% when patients with a low viral load (<1 × 10⁶ copies/mL) were treated with ribavirin. However, the same effect was not observed in ribavirin-treated patients with viral loads greater than 1 × 10⁶ copies/mL; therefore, the authors suggested that ribavirin should be administered as soon as possible to counter SFTSV infection [72]. Other studies have shown no differences in fatality rates between ribavirin-treated and control patients, and ribavirin exerted no significant effect in terms of viral load reduction and recovery of thrombocytopenia regardless of the severity of the disease [90,91,92]. Moreover, most of the studies reporting the effectiveness of ribavirin were conducted using a combination of other treatment options. Ribavirin is also known to cause side effects, such as anemia and hyperamylasemia [48]. Thus, ribavirin administration is not an established effective treatment option [93,94,95].

#### 6.1.2. Favipiravir

Favipiravir (T-705) is a pyrazine derivative and antiviral agent developed in Japan that has a broad spectrum of antiviral effects. Favipiravir inhibits the RNA polymerase of the influenza virus and those of various RNA viruses, such as bunyaviruses, arenaviruses, and the yellow fever virus [96]. Several recent studies have reported the effects of favipiravir on SFTS [97] and suggest that favipiravir has a stronger antiviral effect than ribavirin both in vitro and in vivo. In particular, Tani et al. and Baba et al. reported that favipiravir inhibited the replication of SFTSV in Vero cells, with an IC_50_ of 6 µM and a 50% effective concentration (EC_50_) of 25 µM [98,99]. In addition, Tani et al. reported that the oral administration of favipiravir at dosages of 120 mg/(kg·day) and 200 mg/(kg·day) to interferon receptor 1 knockout mice infected with SFTSV was effective [100]; Gowen et al. showed that stat2 (blocking the initiation of the type I IFN response) knockout hamsters infected with SFTSV were responsive to oral favipiravir treatment at 300 mg/(kg·day), which reduced serum and tissue viral loads [101]. In 2021, Suemori et al. reported a trial on the efficacy and safety of favipiravir [102]. A total of 23 participants who received an oral loading dose of favipiravir 1800 mg twice on the first day and treatment with 800 mg twice a day for a total of 7–14 days tolerated the drug well; the 28-day mortality rate was 17.3%. Adverse effects occurred in approximately 20% of patients. In addition, viral RNA was not detectable in surviving patients at a median of 8 days after favipiravir administration. Because the study was a nonrandomized, uncontrolled, single-arm study, additional randomized clinical trials are required.

### 6.2. Steroids

Severe SFTS causes rapid deterioration, and patients tend to die of MOF, severe bleeding, and septic shock. The key point of this pathogenic process is a cytokine storm. Considering the pathogenesis of SFTS, clinicians envision steroids as a treatment option to suppress the immune system of patients with severe SFTS. This treatment is similar to the role of steroids against the 2019 coronavirus disease (COVID-19) caused by the SARS-CoV-2 virus. Kim et al. reported two cases of successful treatment with a combination of steroids and intravenous immunoglobulin (IVIG) in South Korea [103]. Nakamura et al. reported the effectiveness of steroids in three patients with SFTS combined with encephalopathy in Japan [104]; however, these are all case reports. The autopsy of two patients who died of SFTS in Japan and had been treated with corticosteroids revealed evidence of marked fungal infections in the lungs of both patients. Hiraki et al. suggested that physicians should be aware of possible fungal infections in patients with SFTS, as the disease may suppress cellular immunity [105]. Kato et al. reported fungal infections, such as invasive aspergillosis and oral candidiasis, in approximately 10% (5/48) of patients with SFTS; among the cases of proven fungal infection, 80% (4/5) of patients used corticosteroids [71]. Chen et al. analyzed four cases of SFTS with IPA [106]. Dexamethasone was administered to three of the four patients who exhibited fungal infections in their airways by bronchoscopy. Finally, the three patients were confirmed to have IPA based on histologic, serologic, and radiologic evidence. The other patient had IPA. All patients worsened very quickly, presented with dyspnea, and died due to type 2 respiratory failure. Skakaguchi et al. also reported a case of SFTS with pseudomembranous Aspergillus tracheobronchitis [107]. Patients with SFTS and fungal infection exhibited no common risk factors for invasive aspergillosis. However, considering these case reports, we assume that SFTS can be a risk factor for IPA, even in patients without any apparent risk factor for invasive aspergillosis. Jung et al. evaluated the effects of steroid therapy in patients with SFTS [79] and reported a significant difference in mean 30-day survival between the steroid and nonsteroid groups of mild SFTS patients that was characterized by an APACHE II score of less than 14 points after propensity matching. Moreover, survival differences between the nonsteroid and the early and late steroid groups were statistically significant. However, in patients with severe cases of the disease (initial APACHE II score ≥ 14), 30-day survival did not differ significantly between the steroid and nonsteroid groups after propensity matching. In addition, the authors examined the effects of steroids on 30-day mortality in patients with SFTS using Cox proportional multivariable regression analysis. Steroid-treated patients showed a 3.45-times higher 30-day mortality risk than those who did not receive steroid treatment (Table 1). Therefore, steroid therapy should be used with caution because it increases the occurrence of complications when applied within 5 days from the onset of symptoms or to patients with mild cases of the disease (APACHE II score < 14).

### 6.3. Intravenous Immunoglobulin

IVIG plays an important role in the treatment of various viral diseases by triggering complement activation, virus neutralization, antibody-dependent cellular cytotoxicity, and opsonization. In addition, decreased levels of CD3 and CD4 T cells and increased natural killer T (NKT) cell activity are observed in patients with SFTS [51]. Decreased CD4 T-cell levels are related to reduced immunoglobulin levels, leading to immunosuppression. In addition, the cytokine storm induced by increased NKT cell activity is associated with increased levels of IFN-γ, IL-10, and G-CSF, which are major contributors to SFTS pathogenesis. Based on this knowledge, IVIG administration is believed to help to reduce viral loads in SFTS patients, to contain the spread of the SFTSV, and to effectively suppress the cytokine storm. A case report indicates that patients with serious SFTS were successfully treated with IVIG. Kim et al. and Denic et al. reported cases of severe SFTS patients with complications in which combined treatment with IVIG and other drugs showed positive therapeutic effects [103,108]. However, the effect of IVIG on SFTS remains unclear. In general, IgG antibodies against the measles virus or the varicella zoster virus can be detected in IVIG pharmaceuticals, which are antibodies collected from healthy individuals in the general population. However, neutralization of the SFTSV by IVIG via the abovementioned mechanism may differ from that of measles and varicella zoster viruses because it is estimated that very few antibodies against SFTSV are carried in the general population. Therefore, further studies are required to demonstrate the effectiveness of IVIG treatment.

### 6.4. Plasma Exchange or Convalescent Plasma Therapy

In patients with SFTS, cytokines such as IL-1β, IL-8, macrophage inflammatory protein 1β, and IFN-γ are involved in disease progression and severity [46,108,109]. Thus, plasma exchange could be a possible rescue therapy for cytokine removal. Yoo et al. reported that of 27 patients diagnosed with SFTS from May 2013 to July 2015, clinical manifestations and laboratory findings were improved in 14 patients treated with plasma exchange alone compared with those of 9 patients who did not receive plasma exchange [110]. Similarly, Oh et al. reported that of 53 SFTS patients treated at 9 hospitals in Korea from May 2013 to August 2015, the group that received plasma exchange within 7 days of the onset of symptoms survived longer than the group that did not receive plasma exchange [111].

Choi et al. provided convalescent plasma therapy to patients with SFTS who showed no clinical improvement after early plasma exchange [112]. As a result, patients exhibited decreased serum viral load and an improved clinical course after convalescent plasma therapy. The authors suggested that plasma exchange may play a role in the cytokine storm subsiding during the early phase of the disease, and that convalescent plasma therapy may be applied as a salvage therapy to reduce the viral load in the late phase of the disease. Despite these findings, establishing plasma exchange or convalescent plasma therapy as standard treatment strategies for SFTS is difficult, because the inclusion criteria for patients who can receive plasma exchange are not clear; additionally, the therapeutic effect of plasma exchange or convalescent plasma therapy is not constant. Therefore, acknowledging plasma exchange or convalescent plasma therapy as an established treatment strategy would require a randomized controlled study.

### 6.5. Monoclonal Antibodies

Among several treatment options, monoclonal antibodies are considered new therapeutic agents for SFTS. Guo et al. reported that monoclonal antibodies exerted neutralizing activity against SFTSV infection in Vero cells by binding a linear epitope in the ectodomain of the glycoprotein Gn [113]. This neutralization activity is due to the inhibition of viral cell attachment through the blockade of interactions between glycoprotein Gn and cellular receptors. Kim et al. also reported that their designated antibody was reactive to the envelope glycoprotein Gn of the SFTSV and protected 80% of mice and host cells, suggesting that monoclonal antibodies could protect against SFTSV [114]. These data suggest that monoclonal antibodies could be used as a promising therapeutic option against SFTS.

## 7. Prevention

To date, there have been no commercially available vaccines or chemophylactic agents for the prevention of SFTS. Studies by Reece et al. and Robles et al. showed the possibility of preventing SFTS using vaccines; however, the studies were conducted in animal models [115,116]. In addition, Kwak et al. reported the immunogenic potential and protective efficacy of an SFTSV DNA vaccine [117], which encoded all SFTSV proteins and elicited both neutralizing antibody responses and SFTSV-specific T-cell responses in mice and ferrets. Consequently, vaccinated ferrets were completely protected from lethal SFTSV and did not present with any clinical signs. In addition, the authors used serum transfer studies to confirm that the glycoproteins Gn and Gc are the most effective antigens for inducing protective immunity; additionally, non-envelope-specific T-cell responses contribute to protection against SFTSV infection. Based on these results, Kwak et al. suggested that anti-envelope antibodies play an important role in protective immunity. This study has fostered hope for the future design of preventive vaccines against SFTS. Moreover, Yu et al. demonstrated the immunogenicity and prophylactic efficacy of attenuated recombinant SFTSV capable of triggering humoral immunity in immunized ferrets using two attenuated recombinant SFTSVs (rHB29NSsP102A and rHB2912aaNS) that contained a short 12 amino acid sequence derived from the NS open reading frame and a point mutation at position 102 that induced a proline-to-alanine substitution, respectively [118]. The study suggests that a live attenuated vaccine platform could be suitable for producing vaccine candidates against SFTSV.

Most importantly, the major precaution in endemic areas is to avoid tick bites. After outdoor activities, bedding and clothing should be thoroughly shaken and washed. Other effective prevention methods include taking a shower or bathing immediately after outdoor activities, checking for tick bites, and removing ticks as early as possible. In addition, direct contact with the bodily fluids of domestic animals (e.g., cats or dogs) should be avoided in endemic areas [38,119]. Standard precautions are recommended in SFTS patients; however, because accurately identifying the SFTSV transmission route is difficult, SFTSV could be transmitted via multiple routes in patients with severe SFTS. Therefore, active caution, such as isolation, is required. Healthcare workers should implement standard precautions, including hand hygiene and use of gloves, gowns, goggles, and N-95 face masks while caring for patients with suspected SFTS; the patient room should be disinfected after discharge [120].

## 8. Conclusions

SFTS is a tickborne infectious disease caused by SFTSV that has seen increased case reports year by year (especially in East Asia) and is a serious public health problem. SFTS is accompanied by fever and gastrointestinal symptoms; white blood cells and platelet reduction are the principal laboratory findings and progress rapidly, leading to multiple organ failure, bleeding, and death. Several treatment strategies have been applied to patients with severe SFTS, which is associated with a high mortality rate. However, no effective treatment strategy has yet been established, and patients have mostly received only supportive care. Rapidly suspecting and diagnosing SFTS based on clinical features, laboratory findings, and appropriate epidemiological investigations at the time of patient admission is important for patient survival. In addition, large-scale randomized controlled studies are needed to establish more specific treatment strategies and effective prevention measures to reduce case fatality rates. Finally, more research on drug repositioning for the treatment of SFTS is required, similar to current studies related to COVID-19.

## Figures and Tables

**Figure 1 viruses-13-01213-f001:**
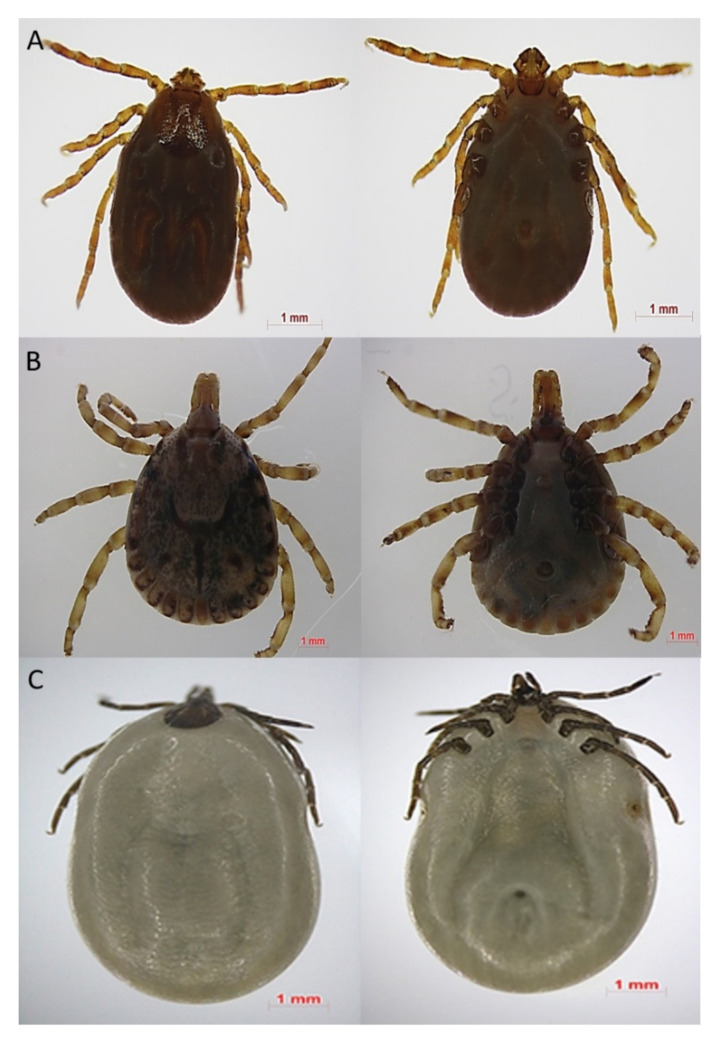
Morphological identification of various tick species in South Korea. (**A**) *Haemaphysalis longicornis,* which is known as a primary vector for SFTSV in SFTS endemic regions; (**B**) *Amblyomma testudinarium,* which can be a potential vector for SFTSV in SFTS endemic regions; (**C**) *Ixodes nipponensis* which can be a potential vector for SFTSV in SFTS endemic regions.

**Table 1 viruses-13-01213-t001:** Treatment effect on 30-day mortality in patients with SFTS [79] *.

Variables	Univariate			Multivariate		
	HR	95% CI	*p*-Value	aHR^†^	95% CI	*p*-Value
Prior antibiotic treatment	1.55	(0.76–3.16)	0.234	1.90	(0.75–4.81)	0.174
Ribavirin	1.61	(0.75–3.45)	0.217	1.06	(0.34–3.25)	0.923
Steroids	4.57	(1.96–10.66)	<0.001	3.31	(1·26–8.73)	0.016
IVIG	1.61	(0.74–3.51)	0.235	0.74	(0.32–1.72)	0.482
Plasmapheresis	2.19	(1.03–4.68)	0.043	1.40	(0.41–4.78)	0.593

Abbreviations: aHR, adjusted hazard ratio; CI, confidence interval; HR, hazard ratio; IVIG, intravenous immunoglobulin. ^†^ Adjusted variables: initial APCHE II score and symptom onset to admission within 7 days. * Copyright permission about Table 1 was obtained from Plos NTD.

## Data Availability

Not applicable.

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
