# Peer review of "Clinical Update of Severe Fever with Thrombocytopenia Syndrome"

_viruses, 2021, doi:10.3390/v13071213_

Round 1

Reviewer 1 Report

Manuscript ID: viruses-1234722

The reviewed article collected clinical updates on the Severe Fever with Thrombocytopenia Syndrome (SFTS). Special focuses were put on the epidemiology, clinical manifestations, laboratory diagnosis, and risk factors of SFTS, as well as the various treatment strategies. This is a rather completed review of SFTS, covering multiple aspects interested to the scientific fields as well as the general publics. My suggestions are listed below.

  1. The last sentence summarizing the skeleton of the review should better reflect the actual subtitles (1-8) in the text.
  2. On the epidemiology, are there SFTS related reports in countries other than the 6 in Eastern Asia? Based on the vector ticks, what are possible areas (countries) the disease/virus might be seen?
  3. The Vietnam positive report was a retrospective serum screening, which may not reflect the actual existence of the virus in the country (the patient might get the disease outside of Vietnam), the description regarding the Vietnam case should be rephrased.
  4. In Taiwan, in addition to the reported human case, there was an earlier report confirming the isolation of SFTSV in the animal (Lin et al., Emerg Microbes Infect, 2020), which adds up to a stronger evidence of the disease occurance in Taiwan.  
  5. Lines 91-98. The tick Rhipicephalus microplus is also a possible vector and should be included in the review. It might be good to slightly expand the vector section.
  6. Please check through the manuscript the correct use of abbreviations. There is no need for abbreviation if the term is used only once or just few times. Some abbreviations are left without preceding full name or even any identification.
  7. Line 226, please check English use.
  8. In the Diagnosis of SFTS (5), it may be of interest to review and compare published PCR methods (to some detail) to aid a more accurate and fast identification of the virus in other countries.
  9. Line 260. By the abstract, one would expect to see a subtitle and contents about “Risk factors“ here.
  10. I feel odd why table and figures from reference 75 is single extensively copied to the current review. In my opinion they are presented out of reasonable proportion in the review. I would suggest if tables and figures were to be used, a summary table or diagram of main contents or related information” for each subtitle” is more deserving.
  11. Line 458. The conclusion (8) should be a conclusion for the overall review rather than just for the “treatment strategy”.

Author Response

Reviewer 1

Manuscript ID: viruses-1234722

The reviewed article collected clinical updates on the Severe Fever with Thrombocytopenia Syndrome (SFTS). Special focuses were put on the epidemiology, clinical manifestations, laboratory diagnosis, and risk factors of SFTS, as well as the various treatment strategies. This is a rather completed review of SFTS, covering multiple aspects interested to the scientific fields as well as the general publics. My suggestions are listed below.

  1. The last sentence summarizing the skeleton of the review should better reflect the actual subtitles (1-8) in the text.

à Answer : The correction is made as you pointed out (Line 474-487)

  1. On the epidemiology, are there SFTS related reports in countries other than the 6 in Eastern Asia? Based on the vector ticks, what are possible areas (countries) the disease/virus might be seen?

à Answer : In addition to the six East Asian countries mentioned, SFTS-like two cases had also been reported in the U.S. We added to this report. (Line 68-74, Reference 20, 21)

à Answer : We described areas where SFTS may occur based on vector ticks (Line 99-102, Reference 30)

  1. The Vietnam positive report was a retrospective serum screening, which may not reflect the actual existence of the virus in the country (the patient might get the disease outside of Vietnam), the description regarding the Vietnam case should be rephrased.

à Answer : The correction is made as you pointed out (Line 64-66)

  1. In Taiwan, in addition to the reported human case, there was an earlier report confirming the isolation of SFTSV in the animal (Lin et al., Emerg Microbes Infect, 2020), which adds up to a stronger evidence of the disease occurance in Taiwan.

à Answer : The correction is made as you pointed out (Line 66-68, Reference 19)

  1. Lines 91-98. The tick Rhipicephalus microplus is also a possible vector and should be included in the review. It might be good to slightly expand the vector section.

à Answer : The correction is made as you pointed out (Line 14, 29, 99)

  1. Please check through the manuscript the correct use of abbreviations. There is no need for abbreviation if the term is used only once or just few times. Some abbreviations are left without preceding full name or even any identification.

à Answer : We checked carefully as you mentioned.

  1. Line 226, please check English use.

à Answer : The correction is made as you pointed out (Line 233)

  1. In the Diagnosis of SFTS (5), it may be of interest to review and compare published PCR methods (to some detail) to aid a more accurate and fast identification of the virus in other countries.

à Answer : The correction is made as you pointed out (Line 243-253, Reference 76,83,84)

  1. Line 260. By the abstract, one would expect to see a subtitle and contents about “Risk factors“ here.

à Answer : We have already mentioned the content of SFTS' risk factors in the “4. Clinical manifestations” section (Line 203-219), so we have not mentioned duplicates here, and we have deleted the "and risk factors of SFTS" part from Abstract (Line 20).

  1. I feel odd why table and figures from reference 75 is single extensively copied to the current review. In my opinion they are presented out of reasonable proportion in the review. I would suggest if tables and figures were to be used, a summary table or diagram of main contents or related information” for each subtitle” is more deserving.

à Answer : We have summarized and revised Tables and Figures based on your advice. (Table 1 and 2, Figure 1 and 2)

  1. Line 458. The conclusion (8) should be a conclusion for the overall review rather than just for the “treatment strategy”.

à Answer : The correction is made as you pointed out (Line 474-487)

Reviewer 2 Report

Authors have compiled the most recent scientific literature on the clinical aspects of SFTS. I found the review logical and easy to read.  A review on SFTSV was published in Experimental & Molecular medicine May 2021, which focuses more on animal models of the disease, whereas this review focuses on human clinical pathology and the current treatments. This is a difficult disease to treat and there does not appear any good treatments available in the clinic to date.

I have couple minor grammatical suggestions for the authors.  I would consider revising the sentence on lines 40-42.  I do find the word “exploded ticks” quite humorous on line 107 as well. On line 181 and 196 I would recommend a different term for altered mentality. I also would recommend on line 308 to say “favipiravir 1800mg twice”. On line 333 use words fungal infection instead of molds.

Include a description for figure 1 and 2 for the censored lines.

For the lines 455-457.  Does the personal protective equipment (PPE) include respiratory protection such a N-95 face mask?

Author Response

Reviewer 2

Authors have compiled the most recent scientific literature on the clinical aspects of SFTS. I found the review logical and easy to read.  A review on SFTSV was published in Experimental & Molecular medicine May 2021, which focuses more on animal models of the disease, whereas this review focuses on human clinical pathology and the current treatments. This is a difficult disease to treat and there does not appear any good treatments available in the clinic to date.

I have couple minor grammatical suggestions for the authors.

  1. I would consider revising the sentence on lines 40-42.

à Answer : The correction is made as you pointed out (Line 40-41)

  1. I do find the word “exploded ticks” quite humorous on line 107 as well.

à Answer : The correction is made as you pointed out (Line 116)

  1. On line 181 and 196 I would recommend a different term for altered mentality.

à Answer : The correction is made as you pointed out (Line 189, 204)

  1. I also would recommend on line 308 to say “favipiravir 1800mg twice”.

à Answer : The correction is made as you pointed out (Line 323)

  1. On line 333 use words fungal infection instead of molds.

à Answer : The correction is made as you pointed out (Line 346-347)

  1. Include a description for figure 1 and 2 for the censored lines.

à Answer : The correction is made as you pointed out (Figure 2)

  1. For the lines 455-457. Does the personal protective equipment (PPE) include respiratory protection such a N-95 face mask?

à Answer : The correction is made as you pointed out (Line 471)

Round 2

Reviewer 1 Report

The manuscript is improved and my concerns are mostly properly responded except for comment No. 10 on the Tables and figures. Figure 1 has good picture quality, however, it is much more relevant to show ticks that may carry SFTSV. Figure 2 and  Table 1 if were the same as in Ref 79, why is it necessary to copy them instead of just reference them?  A table summarizing information on the current manuscript content or related information requiring a table for clearer presentation would be more deserving. 
